# Acceptability and feasibility of long-term, real-time electronic adherence monitoring of HIV pre-exposure prophylaxis (PrEP) use among young women in Kenya: A mixed methods study

Vallery A. Ogello[1], Bernard Kipkoech Rono[2]*, Kenneth Ngure[3,4], Eric Sedah[2], Nicholas B. Thuo[1], Nicholas Musinguzi[5], Jared M. Baeten[4,6], Elizabeth A. Bukusi[2,4], Nelly R. Mugo[1,4], Jessica E. Haberer[7,8]

1 Center of Clinical Research, Kenya Medical Research Institute, Nairobi, Kenya, 2 Center for microbiology research, Kenya Medical Research Institute, Kisumu, Kenya, 3 Department of Community Health, Jomo Kenyatta University of Agriculture and Technology, Nairobi, Kenya, 4 Department of Global Health, University of Washington, Seattle, Washington, United States of America, 5 Global Health Collaborative, Mbarara, Uganda, 6 Gilead Sciences, Foster City, California, United States of America, 7 Center for Global Health, Massachusetts General Hospital, Boston, Massachusetts, United States of America, 8 Department of Medicine, Harvard Medical School, Boston, Massachusetts, United States of America

☯ These authors contributed equally to this work.
* bkrono@kemri-rctp.org

**Data Availability Statement:** The study protocol (including analysis plan), data dictionary, and

## Abstract

Real-time electronic adherence monitoring involves "smart" pill boxes that record and monitor openings as a proxy for pill taking and may be useful in understanding and supporting PrEP use; however, acceptability and/or feasibility for PrEP users is uncertain. We sought to understand the experiences of using a real-time electronic adherence monitor for PrEP delivery among young women in Kisumu and Thika, Kenya. We used the Wisepill device to monitor PrEP use among 18-24-year-old women for two years. Half of the participants were randomized to also receive SMS adherence reminders (daily or as needed for missed doses). We assessed acceptability quantitatively and qualitatively according to the four constructs of Unified Theory of Acceptance and Use of Technology (UTAUT): performance expectancy, effort expectancy, social influence, and facilitating conditions. We assessed feasibility by monitor functionality during periods of PrEP use. We analyzed quantitative data descriptively and compared by site and over time; qualitative data were analyzed inductively and deductively. The median age was 21 years (IQR 19–22), median education was 12 years (IQR 10–13), 182 (53%) had disclosed PrEP use, and 55 (16%) reported recent intimate partner violence. Most participants reported high levels of usefulness and high interest in using the monitor with few problems or worries reported throughout follow-up. Feasibility was high overall with some differences by site (96% functional monitor days in Kisumu vs 88% in Thika). Few monitors were reported lost (N = 29; 8%) or dysfunctional (N = 11; 3%). In qualitative interviews, electronic monitoring was perceived as useful because it supported privacy, confidentiality, easy storage, and PrEP adherence. Effort was

individual participant data that underlie the
quantitative results reported in this Article have
been de-identified and posted on the Harvard
Dataverse. Because qualitative data cannot be fully
de-identified, only the qualitative codebook has
been posted on the Harvard Dataverse in the link
https://dataverse.harvard.edu/dataset.xhtml?
persistentId=doi:10.7910/DVN/PPQKSW..
Transcript data may be shared through appropriate
data use agreements by contacting the Senior
Program Manager for Research at the
Massachusetts General Hospital Center for Global
Health (legarrison@mgh.harvard.edu). The non-
author, institutional point of contact for data access
regarding this paper is legarrison@mgh.harvard.
edu. This email goes to the Senior Program for
Research at the Massachusetts General Hospital
Center for Global Health. Ms Garrison oversees all
regulatory matters for our research group including
data access."

**Funding:** YES, This study was funded by the US
National Institute of Mental Health
(R01MH109309) that was received by JEH and
JMB. The funder had no role in the study design,
data collection and analysis, decision to publish, or
preparation of the manuscript.

**Competing interests:** I have read the journal's
policy and the authors of this manuscript have the
following competing;JMB is currently an employee
of Gilead Sciences, outside of the present work.
JEH reports consultation fees from Merck and
stock ownership in Natera. KN has received
research funds from Merck (MSD) and speaker
fees from Gilead. EAB reports scientific advisory
engagement fees and research support from
Merck(MSD). The other authors declare no
conflicts of interest. This does not alter our
adherence to PLOS ONE policies on sharing data
and materials.

generally considered low. Participants expressed some concern for stigma from monitor
and/or PrEP use. Facilitating conditions involved the monitor size, color, and battery life.
Overall, real-time electronic adherence monitoring was a highly acceptable and feasible
approach to understand PrEP adherence among young women in a sub-Saharan African
setting.

## Introduction

Oral pre-exposure prophylaxis (PrEP) with emtricitabine/tenofovir disoproxil fumarate (FTC/TDF) is >90% effective in preventing acquisition of human immunodeficiency virus (HIV) infection when taken regularly [1, 2]. However, multiple studies have shown that the majority of adolescent girls and young women (AGYW) struggle with daily medication adherence and/or do not persist with PrEP use [3–5]. Medication adherence challenges may be ascribed to multi-level factors. For example, individual factors include low risk perception and depression, while the medication itself may present barriers (e.g., drug side effects and frequency of administration). Social influences include stigma, concerns about disclosure, relationship difficulties (e.g., being perceived as having HIV and/or other sexual partners) as well as complexities of life (e.g., logistical concerns of getting to clinic) [6–9]. Objective, accurate information about adherence behavior is needed to design and assess strategies to help overcome these barriers.

Several approaches have been studied to measure daily pill adherence, although each has limitations. Pill counts, for instance, may be subject to manipulation (e.g., pill dumping). Self-report has been shown to often be inaccurate due to social desirability and recall bias [10, 11], while pharmacy refill only provides the maximum predicted adherence [12]. Pharmacologic measures indicate objective tenofovir (TFV) levels in hair, dried blood spots (DBS), and urine, but these metrics only show cumulative or immediate adherence [13]. None of these measures readily assess patterns of adherence, which can provide important insights into adherence behavior [14]. In particular, patterns of adherence are critical for determining if adherence to PrEP aligns with an individual's risk for HIV acquisition (i.e., prevention-effective adherence) [15].

Electronic monitoring involves a "smart" pill container that records a date-and-time stamp for each opening as a proxy for medication ingestion. Standard electronic monitors, such as the Medication Event Monitoring System (or MEMS), require individuals to return to a clinic to download the data and act on the measurement [16], which typically occurs infrequently; opportunities to support intervention may therefore be missed. Real-time electronic adherence monitors similarly record openings, but transmit the data over cellular networks, thus enabling real-time intervention such as SMS reminders [17]. Although some inaccuracy may arise from monitor non-use or removal of multiple pills at a time, electronic monitors have been shown to be highly acceptable with antiretroviral therapy (ART) and improved adherence when combined with SMS reminders in Uganda [18]. These monitors have also worked well with medications for other conditions, such as diabetes [19]. However, concerns of electronic monitoring have been reported among some populations, such as men who have sex with men (MSM) taking PrEP, including stigma, privacy and portability [20]. Experiences and impact of real-time adherence monitoring may thus differ for PrEP users given differences in perceptions, behaviors, and circumstances. Understanding acceptability and feasibility of real-time electronic adherence monitoring is therefore essential among populations at high risk of HIV acquisition and taking PrEP.

The Unified Theory of Acceptance and Use of Technology (UTAUT) is a model that incorporates well-recognized issues with technology acceptance, as derived from behavioral science and health services literature. Key factors include technology uptake and use (e.g., monitor features) as well as potential challenges seen in particular settings (e.g. stigma) [21]. The UTAUT is based on four constructs: performance expectancy, effort expectancy, social influence and facilitating conditions that can be applied to both qualitative and quantitative assessments of intervention acceptability and feasibility. The overall goal of the model is to understand and explain intentions and challenges to use of technology.

A recent study involved the use of electronic adherence monitoring among young women at high risk of HIV infection in Kenya [22]. Using a mixed method approach, the present analysis sought to understand the acceptability and feasibility of real-time long-term electronic adherence monitoring of HIV PrEP use as informed by the UTAUT in that study.

## Methods

### Study design

The Monitoring PrEP Adherence among Young Adult Women (MPYA) Study has been described in detail elsewhere [22, 23]. Briefly, it was an open-label randomized controlled trial of SMS reminders to support PrEP adherence that was conducted from December 2016 to March 2020 among Kenyan women between 18–24 years of age at high risk of HIV infection [22]. Participants were followed for two years with adherence measured by real-time electronic monitors. PrEP adherence declined over time, and no difference was seen between participants receiving or not receiving the SMS reminders. HIV risk was inconsistently associated with adherence, and low HIV incidence in the study suggested that participants may have achieved protection through multiple strategies, such as changes in sexual behavior and number and type of sexual partners [24].

### Study setting and subject selection

The MPYA Study took place in two locations: Kisumu in western Kenya and Thika in central Kenya. Kisumu is a rural area with high HIV prevalence (17.4%), while Thika is a peri-urban area with HIV prevalence of (5.9%) [25]. The study used community-based strategies to identify young women at risk of HIV infection including partnership with healthcare providers and HIV testing counselors. Participants were eligible if their VOICE risk score was ≥5, which has been associated with an incidence of more than 5 HIV infections per 100 person-years [26]. Women were also eligible if they were in a sero-different relationship (either their sexual partner was known or suspected to be living with HIV). Additional inclusion criteria were being clinically appropriate to start PrEP, sexually active within the previous 3 months, owning a personal mobile phone and being able to send a SMS message, and intending to stay in the local area for at least 1 year. Pregnancy and breastfeeding were exclusion criteria for entry into the study (reflecting evolving data at the time of study initiation on PrEP safety in those contexts), but those who became pregnant during follow-up had the option to continue PrEP. Participants were excluded from the study if they were unable to consent or had a concurrent or prior participation in another study that would influence adherence to PrEP.

### Study procedures

Study participants received a wireless real-time electronic adherence monitor (Wisepill Technologies, South Africa; see Fig 1) at enrolment and were randomized (1:1) to either receive daily SMS reminders or not. The monitor is an internet-enabled medication dispenser that

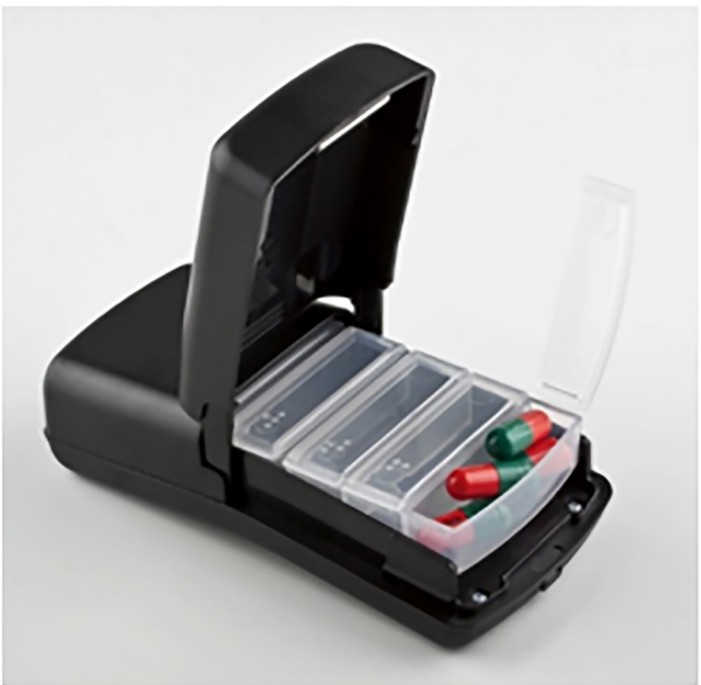

**Fig 1. Wisepill device (116mm lengthx52mm widthx15mm depth).**

allows remote real-time medication management. It holds approximately 30 PrEP tablets and every time it is opened, it records each opening with a date-and-time stamp as a proxy for pill ingestion; this data is then transmitted in real-time to a central server for analysis (with a blinking light indicating data transmission). The monitor has a battery life of up to six months and sends signal reports on the remaining battery power and strength of the transmission signal. The monitors also transmit a daily "heartbeat" to indicate functionality regardless of opening events. Participants in the intervention arm had the option to switch from daily SMS reminders to SMS reminders triggered by the lack of a monitor opening within 30 minutes of the expected dosing time. Study staff trained participants on the function and use of the electronic adherence monitor during study enrolment on how to open and close the device and refilling and arranging drugs in the monitor. Study staff encouraged participants to open it once for each dose and asked them to bring their monitors to each study visit for battery charging or replacement. Those who reported limited cellular network availability in their homes were advised to occasionally access network elsewhere for signals to be sent in the clinic. Participants were asked to use the adherence monitor throughout their PrEP use and report any loss or theft of the monitor for replacement. Staff remotely tracked monitors without heartbeat signal weekly and reached out to participants for further assessment of the monitor when needed.

Study materials were in English, Swahili, or Dholuo depending on each participant's preference. We collected quantitative participant characteristics and socio-behavioural assessments at enrolment, month 1, and every three months through month 24 and entered into REDCap [27] with ongoing quality control checks. Data collection included questions on the acceptability of the electronic adherence monitors with Likert responses (Table 4); questions involved perceived usefulness, complexity, worries about privacy and monitor storage, and interest in monitor use. We assessed feasibility by recording the number of monitors sending the daily

"heartbeat", as well as the number of monitors that were not brought back in time to keep the battery adequately charged.

We also performed serial qualitative interviews among a subset of 50 women at one to two weeks after enrolment, month 3, and month 12. Methods are described in detail elsewhere [24]. Briefly, we purposively selected women for the interviews to reflect age ranges of 18–21 and 22–24 years. Participants who did not return for the follow-up interviews were replaced at month 3 and month 12 at the Kisumu site only. We conducted interviews face-to-face using semi-structured interview guides that included in-depth questions about the acceptability of the electronic adherence monitors according to the UTAUT [21]. Experienced male and female bachelor-level social scientists (authors VO and NBT) conducted the interviews in a language of participants preference in a private room at the research site. We recorded interviews using a digital voice recorder and uploaded them in a password-protected computer. The audio files were transcribed verbatim and translated to English simultaneously where necessary. Verbal participant validation of the information was done during the interviews, and all interview transcripts reviewed by sites' qualitative teams to check for accuracy.

## Data analysis

We used descriptive statistics to summarize the participants' characteristics and responses on the acceptability and feasibility of the electronic adherence monitors. We used the Cochran-Armitage trend test to assess for changes over time, as perceptions may evolve with ongoing use and familiarity. We used Fisher's exact test to compare findings between the study sites given potential differences in social influences and facilitating conditions in these two settings. We limited the latter comparative analysis to the highest or lowest reported response (e.g., "very useful" or "a lot of problems") to facilitate interpretability of any identified differences.

For the qualitative analysis, we used both inductive and deductive content analytic approaches informed by the UTAUT to identify key themes related the acceptability and feasibility of the monitors from the transcripts. Coding was supported by Dedoose software (https://www.dedoose.com/). The research team discussed discrepancies during the first stage of the coding process until a consensus was reached. Analysts (authors VO, NBT, and KN) read through all the transcripts, and VO and NBT coded the interviews using an agreed upon codebook. The data were categorized in broader themes and sub-themes to understand the acceptability and feasibility of the electronic adherence monitor in this context. We analysed concepts that described benefits and usefulness as well as challenges and concerns of the monitor. Our qualitative methods adhered to the COREQ guidelines [28].

## Ethical considerations

This study was approved by institutional review boards at the Kenya Medical Research Institute, University of Washington, and Massachusetts General Hospital. All participants provided written informed consent.

## Results

### Participant characteristics

A total of 348 young women participated in the study. The median age at enrollment was 21 years (interquartile range [IQR] 19–22), and median years in education were 12 (IQR 10–13). Over half had disclosed PrEP use to someone else (N = 182; 53%) and 55 (16%) participants reported intimate partner violence in prior 12 months. Over 90% of participants reported not being married. As shown in Table 1, statistically significant differences between the study sites

**Table 1. Social demographic characteristics for all MPYA participants at enrollment.** N indicates the number of participants; percentage is shown in the parenthesis unless otherwise noted.

| Factor | Total | Kisumu | Thika | P-value |
|---|---|---|---|---|
| N | 348 | 174 | 174 | |
| Median age (IQR) | 21 (19, 22) | 20 (19, 22) | 21 (20, 22) | 0.06 |
| Marital status | | | | 0.62 |
| Not Married | 324 (93) | 161 (92) | 163 (94) | |
| Married (1 partner) | 20 (6) | 12 (7) | 8 (5) | |
| Married (>1 partner) | 3 (1) | 1 (1) | 2 (1) | |
| Education, median years (IQR) | 12 (10,13) | 12 (9,12) | 12 (10,13) | <0.01 |
| Possible depression[1] | 22 (6) | 17 (10) | 5 (3) | 0.01 |
| Travel to clinic >1 hour | 232 (67) | 93 (53) | 139 (80) | <0.001 |
| Employed with a salary | 11 (3) | 4 (2) | 7 (4) | 0.38 |
| Problem alcohol use[2] | 112 (32) | 43 (25) | 69 (40) | <0.01 |
| Disclosed PrEP use | 187 (54) | 122 (70) | 65 (38) | <0.001 |
| Intimate partner violence[3] | 55 (16) | 38 (22) | 17 (10) | <0.01 |

[1]Patient Heath Questionnaire-2 [29]: a response of yes to either question is considered as possible depression.

[2] Rapid Alcohol Problems Screen-4 [30]: a response of yes to one or more items is considered problematic alcohol use in the past year.

[3] Modified Conflict Tactics scale [31]: a response of yes to one or more of the three items shows the presence of violence.

were seen with slightly lower education, more possible depression, less travel time to clinic, less problem alcohol use, more PrEP disclosure, and more intimate partner violence in Kisumu compared to Thika.

## Acceptability

Overall, the majority of participants reported consistently high levels of usefulness and interest in the real-time electronic adherence monitors with low levels of perceived problems, complexity, and worry throughout the study. The highest or lowest reported response values for each acceptability question is compared between the two sites at each time point and presented for changes over time in Table 2 with complete responses shown in Table 3. Notably, no participants reported "a lot" of problems after month 3. More participants found the monitors very useful and were very interested in getting the monitor in Kisumu compared to Thika. Perceived problems and perceived complexity were low and decreased over time in the two sites; however, high levels of worry were more common in Thika than in Kisumu among participants at the end of the study. A total of 24 participants switched from daily SMS reminders to those triggered by lack of monitor openings.

## Feasibility

A total of 165 participants in Kisumu (95%) and 121 (70%) in Thika respectively completed 24 months follow-up. Median follow-up time in months was 22 (22, 23) at both study sites. The adherence monitors did not send a signal to indicate functionality on 6% (N = 6773) of days in Kisumu and 18% (N = 21,837) of days in Thika. Among participants using the monitors during follow up, 22 (13%) participants in Kisumu and 63 (36%) participants in Thika never brought them back to the study sites for battery life maintenance. As shown in Table 4, 29 (8%) monitors were reported lost, 22 (6%) damaged, 2 (1%) not charging, and 11 (3%) not sending data during follow-up. Fewer monitors were brought in for charging and more monitors were lost with time. Problems were reported at 4% of study visits.

**Table 2. Differences in acceptability of the electronic adherence monitor by study site at each time point and combined for the two sites compared over time.** N indicates the number of participants; percentage is shown in the parenthesis. The number of participants commenting on the monitor decreased over time due to loss-to-follow-up.

| | Enrollment (N = 348) | | | Month 3 (N = 304) | | | Month 24 (N = 277) | | | Acceptability for both sites | | | |
|---|---|---|---|---|---|---|---|---|---|---|---|---|---|
| | Kisumu n (%) | Thika n (%) | p-value | Kisumu n (%) | Thika n (%) | p-value | Kisumu n (%) | Thika n (%) | p-value | Enrollment n (%) | Month 3 n (%) | Month 24 n (%) | p-value* |
| | 174 (50) | 174 (50) | | 166 (55) | 138 (45) | | 163(59) | 114 (41) | | 348 (100) | 304 (87) | 277 (80) | |
| Monitor very useful | 166 (95) | 126 (72) | <0.001 | 143 (86) | 111 (80) | 0.21 | 145 (89) | 85 (75) | 0.002 | 292 (84) | 254 (84) | 230 (83) | 0.83 |
| A lot of problems with the monitor | 2 (1) | 6 (4) | 0.28 | 0 (0) | 3 (2) | 0.09 | 0 (0) | 0 (0) | — | 8 (2) | 3 (1) | 0 (0) | 0.01 |
| The monitor seems very complicated | 3 (2) | 4 (2) | 0.99 | 4 (2) | 2 (2) | 0.69 | 0 (0) | 1 (1) | 0.41 | 7 (2) | 6 (2) | 1 (0) | 0.08 |
| Very worried someone will see the monitor | 2 (1) | 4 (2) | 0.69 | 5 (3) | 7 (5) | 0.39 | 2 (1) | 6 (5) | 0.07 | 6 (2) | 12 (4) | 8 (3) | 0.42 |
| Worried about having place to store the monitor | 7 (4) | 10 (6) | 0.62 | 10 (6) | 9 (7) | 0.99 | 5 (3) | 17 (15) | <0.001 | 17 (5) | 19 (6) | 22 (8) | 0.29 |
| Very interested in getting the monitor | 159 (91) | 124 (71) | <0.001 | 148 (86) | 109 (79) | 0.02 | 148 (91) | 86 (75) | <0.001 | 283 (81) | 257 (85) | 234 (85) | 0.34 |

*P-value assess trend over time

**Table 3. Complete results from on monitor acceptability.**

| | | Enrollment | Month 3 | Month 24 | P-value* |
|---|---|---|---|---|---|
| How useful is the monitor? | Very useful | 292 (84) | 254 (86) | 230 (83) | 0.974 |
| | Somewhat useful | 31 (9) | 24 (8) | 26 (9) | |
| | Not at all useful | 12 (4) | 10 (3) | 12 (4) | |
| | Don't know | 13 (4) | 9 (3) | 8 (3) | |
| Do you think you will have any problems in using the monitor | A lot of problems | 8 (2) | 3 (1) | 0 (0) | 0.001 |
| | Some problems | 10 (3) | 6 (2) | 13 (5) | |
| | None at all | 308 (89) | 283 (95) | 256 (92) | |
| | Don't know | 22 (6) | 5(2) | 8 (3) | |
| Does the monitor seem complicated? | Not complicated | 320 (92) | 278 (95) | 258 (93) | 0.114 |
| | Somewhat complicated | 16 (5) | 6 (2) | 9 (3) | |
| | Very complicated | 7 (2) | 6 (2) | 1 (0) | |
| | Don't know | 5 (1) | 4 (1) | 9 (3) | |
| Are you worried that anyone will see the monitor? | Not at all | 303 (87) | 256 (86) | 243 (87) | 0.301 |
| | Somewhat worried | 37 (11) | 24 (8) | 22 (8) | |
| | Very worried | 6 (2) | 12 (4) | 8 (3) | |
| | Don't know | 2 (1) | 5 (2) | 5 (2) | |
| Are you worried about having a place to store the monitor? | Yes | 17 (5) | 19 (6) | 22 (8) | 0.150 |
| | No | 331 (95) | 276 (93) | 252 (91) | |
| | Don't know | 0 (0) | 2 (1) | 3 (1) | |
| How do you feel about getting a monitor? | Very interested | 283 (81) | 257 (86) | 234 (85) | <0.001 |
| | No strong feelings | 47 (14) | 25 (8) | 28 (10) | |
| | Not interested | 1 (0) | 16 (5) | 15 (5) | |
| | Don't know | 17 (5) | 0 (0) | 0 (0) | |

*Fisher's exact test

**Table 4. Real-time electronic adherence monitors use and problems over time.** Values reflect N (%).

| | Month 1 | Month 3 | Month 6 | Month 9 | Month 12 | Month 15 | Month 18 | Month 21 | Month 24 | p-value** |
|---|---|---|---|---|---|---|---|---|---|---|
| N self-reporting using the monitor to take PrEP at each visit* | 283 | 250 | 222 | 189 | 165 | 163 | 161 | 150 | 156 | |
| Monitor charged at study visit | 39 (14) | 79 (32) | 123 (55) | 57 (30) | 88 (53) | 62 (38) | 70 (44) | 61 (41) | 43 (28) | <0.001 |
| No problems reported | 275 (97) | 246 (98) | 211 (95) | 181 (96) | 158 (96) | 159 (98) | 156 (97) | 145 (97) | 145 (93) | 0.10 |
| Problems reported**** | | | | | | | | | | |
| Lost | 1 (0.5) | 2 (1) | 5 (2) | 3 (1.5) | 4 (2) | 2 (1) | 1 (0.6) | 3 (2) | 8 (5) | <0.001 |
| Not sending data | 1 (0.5) | 2 (1) | 2 (1) | 2 (1) | 1 (0.6) | 1 (0.6) | 0 (0) | 1 (0.7) | 1 (0.6) | 0.79 |
| Faulty battery | 0 (0) | 0 (0) | 0 (0) | 0 (0) | 1 (0.6) | 0 (0) | 0 (0) | 0 (0) | 1 (0.6) | 0.17 |
| Other*** | 6 (2) | 0 (0) | 4 (2) | 3 (1.5) | 1(0.6) | 1(0.6) | 4 (2.5) | 1 (0.7) | 2 (1) | 0.86 |

*Participants who missed the visits or were on a study drug hold did not provide this data.

**P-value assess the trend over time

***Burnt, damaged by partner, monitor accidentally swapped between participants, thrown away, low battery

**** A participants may report multiple problems

## Qualitative findings

*Overview.* Fifty participants were interviewed at enrolment (serial interview 1), 43 at month 3 (serial interview 2) and 42 at month 12 (serial interview 3) for a total of 135 interviews. The interviews lasted an average of 43 minutes. Participants described electronic monitoring as useful consistently over time because it supported privacy, confidentiality, easy storage, and PrEP adherence. Effort was generally considered low, shifting over time as participants reported to have learnt how to use the device. Participants expressed some concern for stigma from monitor and/or PrEP use. Facilitating conditions involved the monitor size, color, and battery life, although some were concerned that the monitor light that could lead to unintended disclosure of PrEP use. Qualitative findings are summarized below according to the UTAUT model [21] in Fig 2.

## Perceived usefulness

**Privacy and confidentiality.** The electronic adherence monitor was perceived as useful because it was private and confidential; this perception did not change throughout the study. Participants described the role of the monitor as a device meant to ensure privacy was maintained, especially among those who wanted to be in control over who knew they were using PrEP. The perceived privacy offered by the electronic monitor also protected them against potential stigma related to PrEP use.

*"Like I said, it's really good to keep the medicine in the Wisepill device compared to the container [standard pill bottle] because the container is exposed. But the Wisepill device is a little bit confidential and private because no one can see whatever is inside the box."*

**[Month 3, Kisumu]**

Participants further described that the monitor would protect them from negative perceptions in the community associated with pill taking due to its confidential nature.

*It gives me the confidentiality because at times you may be going somewhere, and people will not know what it is because most of the time people will judge you when they see you take*

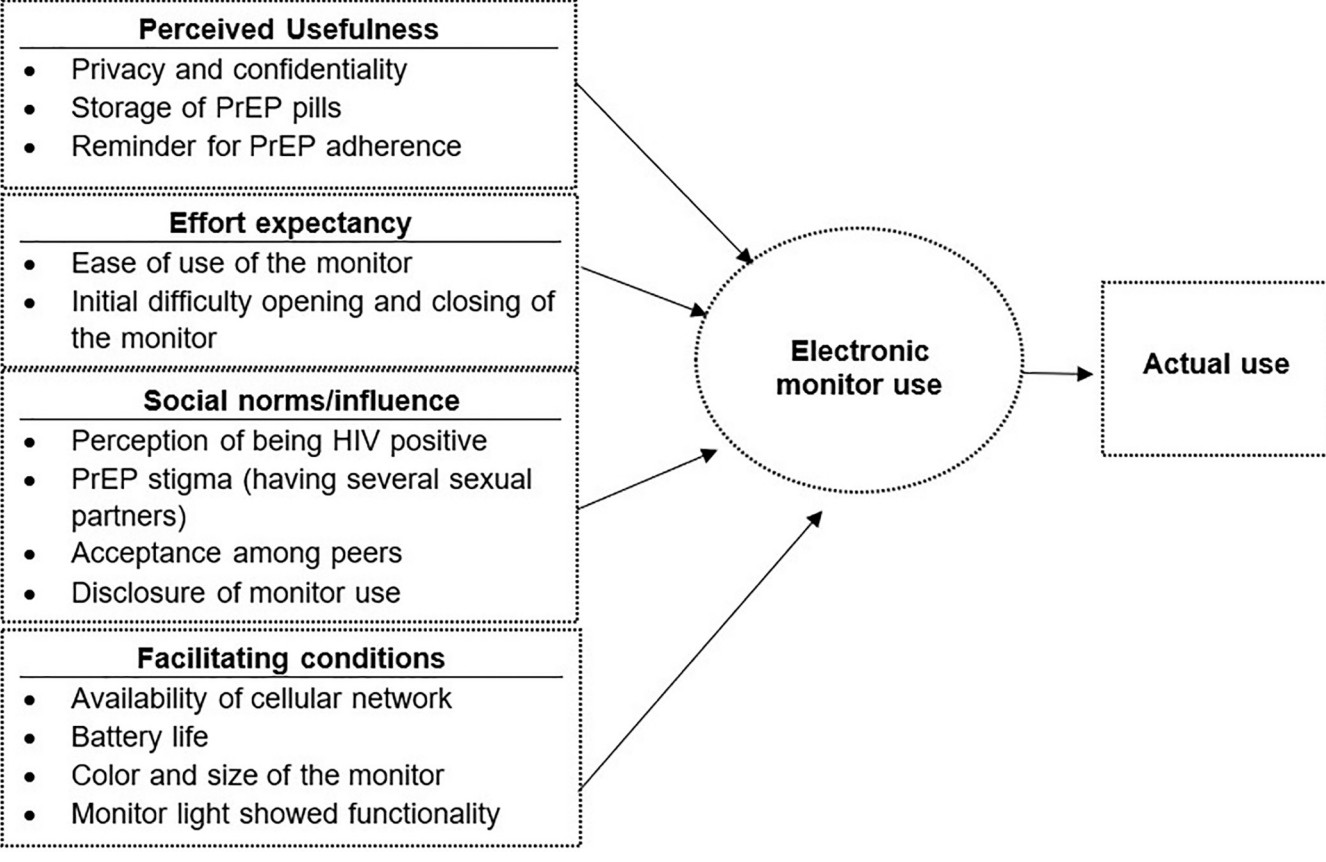

**Fig 2. The Unified Theory Acceptance and Use of Technology (UTAUT) model.**

*drugs all the time. But when I carry the Wisepill device, it's very hard for them to know what is inside there they will think it is something but not for drugs.*

**[Enrollment interview, Kisumu]**

**Storage of PrEP pills.** Participants perceived the electronic adherence monitor as a storage device for the pills (i.e., PrEP) that kept the pills safe from getting 'exposed' or from 'spilling' in comparison to the pill bottle. The portable nature of the monitor was also appealing to the participants, and they felt they could easily travel with it.

*"That container is good. I used to use paper bags, and medicine and that packaging can get spoiled and spill the medicines, but that medicine will not get spoilt when in the Wisepill."*

**[Enrollment Interview, Thika]**

*"Wisepill has helped me in storing the medicines. Again, when I am traveling especially when I will be spending the night there; I'll carry it along with me. I will open, use it and keep it away. No one will know. When you carry the other containers and someone gets them in your bag, that person can get suspicious."*

**[Month 12, Kisumu]**

**Reminder for PrEP adherence.** The electronic monitor was viewed as an adherence reminder. Participants were particularly happy with the fact that 'they were being seen', as the monitor sent a signal to the staff demonstrating their adherence. To them, it was an indication that they were serious with their decision to take PrEP and a motivator to adhere to their medication.

*R: . . .I know if I don't open it, it shows that I have not taken medicine, so it means I have to take the medicine. . . (laughs) not so that it can be seen that I have taken medicine but for my own good. Yah*

*I: when you say for your own good you mean?*

*R: those drugs are supposed to help me so if I refuse to take them, I will be the one at risk*

**[Enrollment interview, Thika]**

*I: Thank you. How has Wise pill device helped you?*

*R: Yes. . .. it has helped me. I said earlier that when you have people around you and want to take PrEP; with the Wise pill you can take your medicines. It also shows a network in the office. This helps because you know that if you do not take it, then it will show. The Study team will think that you have boycotted taking PrEP. It reminds you to take your drug. They can help me in the future knowing that I am in need of those drugs.*

**[Month 3, Kisumu]**

## Effort expectancy

**Ease of use of the electronic adherence monitor.** Several participants reported that it was easy learning how to open and close the monitor. Some participants also mentioned to have initially struggled to open or close the monitor when they were learning how to use it. Very few had persistent challenges of use at month 3 and month 12.

*"I can't say that it's easy to me I open it very well but closing it. . . (Laughs). . .only closing it (laughs). . .closing it is a bit of a challenge but now that I've used it for a while, I'm getting used to it but it's still a challenge."*

**[Enrollment interview, Kisumu]**

*"I only faced challenge with opening it when I started taking PrEP. I do not have any problem with it now."*

**[Month 3 interview, Kisumu]**

## Social norms and influence

**Perception of living with HIV.** Participants reported the potential of being perceived as living with HIV when they are seen with the electronic adherence monitor. The concerns were rather based on the contents of the monitor (i.e., PrEP) and not the monitor itself. Participants said that not everyone knew about electronic monitoring, and others would think that they were taking antiretroviral therapy (ART) for HIV infection because both medications are taken daily.

"*I did not want anyone to see it; this could lead to rumors or misconceptions about me. Someone will think that I am HIV positive. I keep it safely that it is not easy for anyone to see it lest someone accuse me of taking ART.*"

**[Enrollment Interview, Kisumu]**

"*He will wonder what it is. Then when you explain to him that it is for preventing HIV, he will see like you do not trust him or you have other partners.*"

**[Enrollment interview, Thika]**

**PrEP stigma.**    Participants further described the use of the monitor to protect them from potential PrEP stigma. They mentioned that those taking PrEP are sometimes perceived as having several sexual partners and the privacy the monitor offered protected potential stigma.

"*First, it has helped me to take the drugs because many at times those who take PrEP are perceived to have several partners or that your partner is infected, and you are trying to protect yourself. But to me no one can know what is inside if I don't explain it to them.*"

**[Month 12 interview, Kisumu]**

**Acceptance among peers.**    Participants explained that their peers and other social networks perceived the electronic monitoring as an appealing technology that supported medication use. For instance, one young woman mentioned that she was perceived as being 'clever' because she was using the monitor.

"*When I open it. . .. even someone is around me I don't feel ashamed. So, I open it whole heartedly because it is something good. Even when someone sits next to me, he will think that this person is doing something very clever because when I open, he will see the light and once he has seen the light, he will think that . . .Eh. . . (Sigh)This lady is clever because the device has some technology with it.*"

**[Enrollment interview, Kisumu]**

**Disclosure of electronic monitor use.**    Most participants reported to have disclosed electronic adherence monitor use and PrEP use to their partners, friends/peers, or family members. Disclosure of monitoring was not always associated with the contents (i.e., PrEP), and participants who reported non-disclosure of monitoring also reported non-disclosure of PrEP use that was either related to HIV stigma or PrEP use stigma.

"*I: Okay, thank you. What are the challenges related to using the Wisepill?*

*R: (laughs) (silence)*

*I: Maybe we can look at challenges using the device when it comes to relationships*

*R: I just hide it from him (laughs). . .I have no plans of showing him*

*I: Okay. What are the reasons for that?*

*R: I don't want him to know that I am using drugs"*

**[Enrollment interview, Thika]**

*"You know when you inform people, they think that you have HIV. . .I think instead of them thinking it is PrEP, they think you are positive"*

**[Enrollment interview, Thika]**

### Facilitating conditions

**Monitor features and functionality.**   Participants described facilitating conditions especially during follow-up interviews at month 3 and 12 such as the physical appearance of the monitor, the battery life, and the ability of the monitor to work well in good network conditions. The size of the monitor was particularly mentioned to be small and likened to a phone that could easily fit in the purse to allow portability as described by a participant: *"It does not take a big space when carrying, I can carry it in my purse wherever I go".* The black color of the monitor was preferred because participants believed it attracted less attention.

*"Because you can easily hide the black color because if it was white, it would have been shouting. Everyone would want to know what it is".*

**[Month 12, Thika]**

Conversely, a few participants expressed contradictory views on size. Some felt the monitor should be smaller to favor portability, while others felt it should be bigger to increase capacity to hold more pills.

*"I can say that the size is bad. It is quite big especially when you open it. It could have been good if it is a little slimmer something like the spectacle case or something that resembles an office equipment".*

**[Month 12, Kisumu]**

*"Yes, but I would suggest that you expand the size of the container so that it can fit in the medicines well. . . If the container was bigger, it could carry a lot of medicine"*

**(Month 3, Thika)**

The light emitted by the monitor during signal transmission was perceived as an indication that the monitor was functional (e.g., the battery was fully charged when the light was green). This made participants confident and able to contact the clinic whenever they did not see the light.

*"Okay with my wise pill there was a time it was not producing that light and I was told it needed to be charged so I brought it back to the clinic and I was given another one".*

**(Month 12, Kisumu)**

A few participants felt the light attracted attention and were concerned about the possibility of the electronic monitor sending a signal to the clinic whenever someone else opened the monitor out of curiosity. For instance, participants gave examples of the possibility of children

playing with the monitor due to the light. Some felt the lighting was prolonged citing *'it should only blink shortly when opened'*. Some participants felt the light could lead to unintended disclosure of PrEP use.

"*At times it's [light] annoying because like I said not everybody knows that you are using that you are using the Truvada. Maybe you are using it and then you just open it and take out the pill and then somebody else is in the house, and then you lock it and still its blinking wherever you go (laughter)*".

*[Month 3, Kisumu]*

## Discussion

This mixed methods longitudinal study contributes an in-depth understanding of acceptability and feasibility for long-term use of real-time electronic adherence monitors in the context of HIV prevention. Young women reported high levels of acceptability and interest in real-time electronic adherence monitoring while taking PrEP in both study sites. Overall, most participants reported consistent high levels of usefulness; complexity and worry remained low with just a few reporting "a lot" of problems and these reports diminished over time. Feasibility was also high with monitor functionality on the vast majority of days and few monitors with technical problems or loss. In the interviews, participants provided generally positive reviews of the monitors. They highlighted how the monitors promoted privacy and confidentiality, and they found them easy to use. Perceived stigma was focused more on PrEP than the monitor, and most participants were comfortable with monitor disclosure among family and peers. Features of the monitor like size, battery life, availability of network, and color facilitated its use with limited concerns noted.

The high acceptability in our study is similar to other studies using the device for monitoring of ART [18, 32] as well as other treatment medications where no concerns have been reported [19]. Notably, in a qualitative study conducted in Chicago among young African American MSM on ART, electronic monitoring was shown to be acceptable and useful [33] with few concerns of privacy and potential disclosure of HIV status resulting from involvement of close contacts to reinforce adherence. Similarly, a few of our participants reported stigma as a concern with the electronic monitoring that was associated with 'daily pill taking'. On the contrary, electronic monitoring offered privacy to our participants as it concealed PrEP use, hence views on privacy may differ among different populations. In other contexts, for instance in China, electronic monitoring of people taking ART showed acceptability concerns related to inconvenience and worry of HIV status disclosure [34]. The sample size for that study, however, was small (N = 10) with only one month of follow-up. Elsewhere in Uganda, a study among gender diverse sex workers showed poor acceptability of electronic monitor(i.e., device non-use) likely due to stigma [35] In our study, problems diminished over time of two years among those who continued to use PrEP. Concerns related to monitor features, such as its light, color, and size argue for tailor-made adherence monitors to suit different participant preferences and optimize use. Importantly, contexts of acceptability may also differ in treatment and prevention.

Perceived usefulness and interest in the monitor were higher in Kisumu than Thika at all timepoints, while all other aspects of acceptability were similar. Kisumu site reported higher disclosure of PrEP use and less alcohol use compared to the Thika site, which may explain some of the higher reported usefulness and interest with electronic monitor. Further, the prevalence of HIV is higher in Kisumu compared to Thika [25], and new PrEP delivery models

such as community-based interventions (DREAMS) have been implemented to expand access [36]. The high awareness may also contribute to high acceptability of HIV prevention technologies [37].

The real-time electronic adherence monitors were largely feasible in our study. The monitor functionality was documented over 90% of monitored days, although less so in Thika compared to Kisumu. The lower numbers in Thika primarily stem from more than a third of participants not bringing the monitor at follow up visits for battery life maintenance. Other studies have also demonstrated the use of electronic adherence monitor to be feasible for medication monitoring even in resource limited setting [38, 39]. Technical difficulties such as lack of connectivity, network failures, and poor data transmission have been reported as impediments to real-time monitoring of adherence, for example in the rural southern US [40]; however, in our study less than 10% of the monitors were reported lost despite the long follow-up period, and problems such as data transmission and faulty batteries were rare. Our greater success may be due to improvements in the monitors in recent years, including solid state components, upgraded modem, improved network compatibility, and longer battery lives.

The persistent perception of privacy and confidentiality as a valued aspect of the monitors is notable. Stigma related to the daily pill taking behavior is a common challenge among PrEP users and can impair adherence [41, 42]. PrEP stigma concerns have also been reported in other settings with electronic adherence monitor use [20, 35]. We found that perceived social stigma was associated with the contents of the monitor (i.e., PrEP) and not the device itself, hence the potential to deliver rapid adherence support and/or behavioral feedback with electronic monitoring should be explored. Additionally, participants liked the fact that the monitor was portable. In this regard, the monitor facilitated comfort of movement without potential exposure to stigma. Electronic adherence monitors may have also supported PrEP adherence as participants reported that it acted as an adherence reminder to take PrEP. Participants knew that any missed doses could be detected by the study staff and knowing that they were 'being seen' motivated their PrEP use. Because of this, study participants could have potentially opened the wise pill device and fail to take medication without the knowledge of the study staff. This finding is consistent with the use of similar monitors for ART in Uganda, where participants perceived 'being seen' to facilitate ART adherence [18].

Our study had important strengths and limitation. First, the use of mixed method approach helped understand the nature of acceptability and feasibility reported in the quantitative findings. Secondly, the fact that each individual participant was followed over a two-year period had the potential to demonstrate long-term perceptions. The primary limitation of the study is the potential for social desirability bias; the participants' knowledge that they were being monitored which could have triggered device openings that did not correspond to true PrEP adherence (i.e., pill ingestion). Experiences with electronic monitors outside of a supportive research context may also differ.

## Conclusion

In summary, our findings demonstrate that the use of electronic monitoring was highly acceptable and feasible for young women taking PrEP over the course of the study with few reporting concerns and challenges. Electronic monitoring has high value for understanding PrEP adherence among young women and potentially in other populations and settings. Key components of the acceptability included the privacy and confidentiality provided by the monitors, as well as the connection it created with the study staff. These elements could be further explored as mechanisms to support young women and other individuals in adherence to their medication.

## Acknowledgments

We thank the participants in this study, who contributed their time and effort to make the work possible, as well as the entire MPYA team:

*Principal Investigators*: Jessica E. Haberer, Jared M. Baeten, Elizabeth Bukusi, Nelly Mugo.

*Co-Investigators*: Kenneth Ngure, Ruanne Barnabas, Harsha Thirumurthy, Ingrid Katz

*Study team members* (Kisumu): Kevin Kamolloh, Josephine Odoyo, Linda Aswani, Lawrence Juma, Elizabeth Koyo, Bernard Rono, Stanley Cheruiot, Vallery Ogello, Loice Okumu, Violet Kwach, Alfred Obiero, Stella Njuguna, Millicent Faith Akinyi, Lilian Adipo, Sylvia Akinyi

*Study team members* (Thika): Catherine Kiptiness, Nicholas Thuo, Stephen Gakuo Maina, Irene Njeru, Peter Mogere, Sarah Mbaire, Murugi Micheni, Lynda Oluoch, John Njoroge, Snaidah Ongachi, Jacinta Nyokabi

*Program manager*: Lindsey Garrison

*Statisticians/analysts*: Maria Pyra, Katherine K. Thomas, Nicholas Musinguzi, Susie Valenzuela

## Author Contributions

**Conceptualization:** Jared M. Baeten, Elizabeth A. Bukusi, Nelly R. Mugo, Jessica E. Haberer.

**Formal analysis:** Vallery A. Ogello, Nicholas B. Thuo, Nicholas Musinguzi.

**Funding acquisition:** Jared M. Baeten, Jessica E. Haberer.

**Investigation:** Jared M. Baeten, Elizabeth A. Bukusi, Nelly R. Mugo, Jessica E. Haberer.

**Methodology:** Kenneth Ngure, Jared M. Baeten, Elizabeth A. Bukusi, Jessica E. Haberer.

**Supervision:** Kenneth Ngure, Jared M. Baeten, Elizabeth A. Bukusi, Jessica E. Haberer.

**Writing – original draft:** Vallery A. Ogello, Bernard Kipkoech Rono, Jessica E. Haberer.

**Writing – review & editing:** Vallery A. Ogello, Bernard Kipkoech Rono, Kenneth Ngure, Eric Sedah, Nicholas B. Thuo, Nicholas Musinguzi, Jared M. Baeten, Elizabeth A. Bukusi, Nelly R. Mugo, Jessica E. Haberer.

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
