## [Decision Letter · Decision Letter 0]

29 Nov 2023

PONE-D-22-24605Acceptability and Feasibility of Long-Term, Real-Time Electronic Adherence Monitoring of HIV Pre-Exposure Prophylaxis (PrEP) Use among Young Women in Kenya: A Mixed Methods StudyPLOS ONE

Dear Dr. Rono,

Thank you for submitting your manuscript to PLOS ONE. After careful consideration, we feel that it has merit but does not fully meet PLOS ONE’s publication criteria as it currently stands. Therefore, we invite you to submit a revised version of the manuscript that addresses the points raised during the review process.

 Please address the suggestions made by the reviewers- use your best judgement on what should be changed. 

We look forward to receiving your revised manuscript.

Kind regards,

Jill Blumenthal

Academic Editor

PLOS ONE

Journal Requirements:

"I have read the journal's policy and the authors of this manuscript have the following competing;JMB is currently an employee of Gilead Sciences, outside of the present work. JEH reports consultation fees from Merck and stock ownership in Natera. KN has received research funds from Merck (MSD) and speaker fees from Gilead. EAB reports  scientific advisory engagement fees and research support from Merck(MSD). The other authors declare no conflicts of interest."

Additional Editor Comments:

Please address the comments made by the reviewers. Once that is complete, the manuscript can be accepted.

Reviewers' comments:

Reviewer's Responses to Questions

**Comments to the Author**

1. Is the manuscript technically sound, and do the data support the conclusions?

Reviewer #1: Yes

Reviewer #2: Yes

2. Has the statistical analysis been performed appropriately and rigorously? 

Reviewer #1: Yes

Reviewer #2: I Don't Know

3. Have the authors made all data underlying the findings in their manuscript fully available?

Reviewer #1: Yes

Reviewer #2: Yes

4. Is the manuscript presented in an intelligible fashion and written in standard English?

Reviewer #1: Yes

Reviewer #2: Yes

5. Review Comments to the Author

Reviewer #1: Dear Author,

Thank you for this study

It is a very good addition to the previous studies/ research on prevention of HIV.

This is a highly studious work with excellent originality.

Abstract: It is interesting and validly comprises of the right expectation in the body of the study .. Thank you.

Introduction: The social and scientific values of the study was apparent, and the expected bridging function was clear

Methodology: The study design, setting and procedure follow all the necessary rules for the mixed study.

Analysis :Analysis was comprehensive and reasonable. No conflict noticed. The rules guiding the UTATUT in scientific studies were delicately respected.

Discussion: The discussions were original and support with the findings in the study

I concluded that this is a well conducted study, very relevant and specific.

Please find attached : most grammatical errors.

Congratulations

Reviewer #2: This is a well-written and important paper that adds to the literature around adherence monitoring for PrEP by assessing young people aged 18-24 at high risk of HIV in Kenya. The analysis and results were well laid-out, and use of the Unified Theory Acceptance and Use of Technology (UTAUT) model provided a clear structure for reporting of qualitative results (Fig 1). The quotes are informative and lend valuable insight to the quantitative data. I have some minor suggestions/comments below:

- Abstract line 43 is confusing as written "high levels of usefulness interest"; please clarify

- Consider merging Tables 2 and 3 (both Acceptability results) so that the data in Table 2 appears as a 4th column in (current) Table 3 (all sites)

- You mention in Study Procedures pg 6 that participants in the Intervention arm had the option to switch from daily SMS reminders to reminders triggered by lack of monitor opening. Can you include in Results the proportion of participants who opted to do this? It's an interesting and valuable data point in itself.

- Please include a discussion of whether participants could open the bottle but not take the pill, and whether this was assessed as a potential limitation to use of Wisepill devices to accurately measure adherence to PrEP. If this was not addressed at all in this study, please state this as well.

6. PLOS authors have the option to publish the peer review history of their article (what does this mean?). If published, this will include your full peer review and any attached files.

Reviewer #1: **Yes: **Adeloye Amoo Adeniji

Reviewer #2: **Yes: **Helen Koenig

---

## [Author Response · Author response to Decision Letter 0]

20 Jan 2024

Editor’s comments

We thank the Editor for the important guidance. We have ensured that the manuscript meets PLOS ONE's style requirements.

2. Please include a complete in your revised copy of PLOS’ questionnaire on inclusivity in global research manuscript. 

We have included copy of PLOS’ questionnaire on inclusivity in global research. 

"I have read the journal's policy and the authors of this manuscript have the following competing; JMB is currently an employee of Gilead Sciences, outside of the present work. JEH reports consultation fees from Merck and stock ownership in Natera. KN has received research funds from Merck (MSD) and speaker fees from Gilead. EAB reports scientific advisory engagement fees and research support from Merck (MSD). The other authors declare no conflicts of interest." Please confirm that this does not alter your adherence to all PLOS ONE policies on sharing data and materials, by including the following statement: "This does not alter our adherence to PLOS ONE policies on sharing data and materials.”

We have confirmed that the competing interest does not alter our adherence to all PLOS ONE policies on sharing data and materials and have included the following statement: "This does not alter our adherence to PLOS ONE policy on sharing data and materials.”

4. In your Data Availability statement, you have not specified where the minimal data set underlying the results described in your manuscript can be found.

"We have updated the Data Availability statement as follows, but would be happy to see your version if preferable: “The study protocol (including analysis plan), data dictionary, and individual participant data that underlie the quantitative results reported in this Article have been de-identified and posted on the Harvard Dataverse. Because qualitative data cannot be fully de-identified, only the qualitative codebook has been posted on the Harvard Dataverse. Transcript data may be shared through appropriate data use agreements by contacting the Senior Program Manager for Research at the Massachusetts General Hospital Center for Global Health (legarrison@mgh.harvard.edu).”

We thank the Editor for this point. We have moved the ethics statement to the method section of the paper.

Reviewer #1: 

Dear Author,

Thank you for this study

It is a very good addition to the previous studies/ research on prevention of HIV.

This is a highly studious work with excellent originality.

Abstract: It is interesting and validly comprises of the right expectation in the body of the study. Thank you.

Introduction: The social and scientific values of the study was apparent, and the expected bridging function was clear

Methodology: The study design, setting and procedure follow all the necessary rules for the mixed study.

Analysis: Analysis was comprehensive and reasonable. No conflict noticed. The rules guiding the UTATUT in scientific studies were delicately respected.

Discussion: The discussions were original and support with the findings in the study

I concluded that this is a well conducted study, very relevant and specific.

1. Please find attached: most grammatical errors.

We thank the Reviewer for the positive observations about our study. We have taken note and corrected all grammatical errors in the revised version of the paper.

Reviewer #2:

This is a well-written and important paper that adds to the literature around adherence monitoring for PrEP by assessing young people aged 18-24 at high risk of HIV in Kenya. The analysis and results were well laid-out, and use of the Unified Theory Acceptance and Use of Technology (UTAUT) model provided a clear structure for reporting of qualitative results (Fig 1). The quotes are informative and lend valuable insight to the quantitative data. I have some minor suggestions/comments below:

1. Abstract line 43 is confusing as written "high levels of usefulness interest"; please clarify

We have clarified the sentence to read, “Most participants reported high levels of usefulness and high interest in using the monitor with few problems or worries reported throughout follow-up.”

2. Consider merging Tables 2 and 3 (both Acceptability results) so that the data in Table 2 appears as a 4th column in (current) Table 3 (all sites)

Table 2 and Table 3 have been merged as recommended and labelled Table 2.

3. You mention in Study Procedures pg 6 that participants in the Intervention arm had the option to switch from daily SMS reminders to reminders triggered by lack of monitor opening. Can you include in Results the proportion of participants who opted to do this? It's an interesting and valuable data point in itself.

We appreciate this suggestion and have added the following statement on line 242, “A total of 24 participants switched from daily SMS reminders to those triggered by lack of monitor openings.”

4. Please include a discussion of whether participants could open the bottle but not take the pill, and whether this was assessed as a potential limitation to use of Wisepill devices to accurately measure adherence to PrEP. If this was not addressed at all in this study, please state this as well.

We have revised a statement in discussion, line 530, to read, “The primary limitation of the study is the potential for social desirability bias; the participants’ knowledge that they were being monitored could have triggered device openings that did not correspond to true PrEP adherence (i.e., pill ingestion).”

---

## [Editor Report · Decision Letter 1]

6 Feb 2024

Acceptability and Feasibility of Long-Term, Real-Time Electronic Adherence Monitoring of HIV Pre-Exposure Prophylaxis (PrEP) Use among Young Women in Kenya: A Mixed Methods Study

PONE-D-22-24605R1

Dear Dr. Rono,

We’re pleased to inform you that your manuscript has been judged scientifically suitable for publication and will be formally accepted for publication once it meets all outstanding technical requirements.

Kind regards,

Jill Blumenthal

Academic Editor

PLOS ONE

---

## [Editor Report · Acceptance letter]

26 Feb 2024

PONE-D-22-24605R1 

PLOS ONE

Dear Dr. Rono, 

I'm pleased to inform you that your manuscript has been deemed suitable for publication in PLOS ONE. Congratulations! Your manuscript is now being handed over to our production team.

Kind regards, 

on behalf of

Dr. Jill Blumenthal 

Academic Editor

PLOS ONE